# In-Bulk Temperature Profile Mapping Using Fiber Bragg Grating in Fluids

**DOI:** 10.3390/s23208539

**Published:** 2023-10-18

**Authors:** Sylvie Su, Tianyi Niu, Tobias Vogt, Sven Eckert

**Affiliations:** 1Helmholtz-Zentrum Dresden–Rossendorf, Bautzner Landstraße 400, 01328 Dresden, Germany; 2Faculty of Electrical and Computer Engineering, Technische Universität Dresden, 01062 Dresden, Germany

**Keywords:** optical fiber, Fiber Bragg Grating (FBG), temperature sensor, in-bulk measurement, multiplexing, temperature mapping

## Abstract

The capabilities of Fiber Bragg Grating (FBG) sensors to measure temperature variations in the bulk of liquid flows were considered. In the first step of our research project, reported in this paper, we investigated to what extent the use of thin glass fibers without encapsulation, which only minimally disturb a flow, can fulfill the requirements for robustness and measurement accuracy. Experimental tests were performed in a benchmark setup containing 24 FBG measuring positions, which were instrumented in parallel with thermocouples for validation. We suggest a special assembly procedure in which the fiber is placed under a defined tension to improve its stiffness and immobility for certain flow conditions. This approach uses a single FBG sensor as a reference that measures the strain effect in real time, allowing accurate relative temperature measurements to be made at the other FBG sensor points, taking into account an appropriate correction term. Absolute temperature readings can be obtained by installing another well-calibrated, strain-independent thermometer on the reference FBG. We demonstrated this method in two test cases: (i) a temperature gradient with stable density stratification in the liquid metal GaInSn and (ii) the heating of a water column using a local heat source. In these measurements, we succeeded in recording both spatial and temporal changes in the linear temperature distribution along the fiber. We present the corresponding results from the tests and, against this background, we discuss the capabilities and limitations of this measurement technique with respect to the detection of temperature fields in liquid flows.

## 1. Introduction

Reliable knowledge of temperature fields in fluid flows is extremely important to monitor and better understand heat transport in flow processes. Temperature measurements also provide indirect information about the flow structures, flow regimes, and operating conditions in experimental setups or industrial facilities. The motivation for the present study arose from investigations of turbulent thermal convection in liquid metals and the interest in how the resulting flow structures influence the heat transport in the system [1,2,3,4,5]. Both numerical and experimental works have shown that detailed temperature data in the bulk of the fluid provide crucial information about local heat transport, leading to a more accurate and advanced understanding of the studied system [6]. Having access to a temperature map within a layer of turbulent fluid is very attractive, as it allows one to characterize the thermal structures and track their dynamics. Existing experiments for presenting internal thermal structures rely on measurement techniques that are either intrusive or based on optical images, such as shadowgraphy or, more recently, the use of thermochromic crystals [7,8,9]. However, for some applications, typically with opaque fluids such as liquid metals, optical access is limited or impossible. Optical-fiber-based sensors provide an interesting alternative for this problem. Optical fibers are small, thermally stable, and do not carry electrical current; they are therefore particularly attractive for liquid metal applications, which often involve high temperatures, strong magnetic fields, and confined environments. Various methods to obtain temperature information from optical fibers have been developed. Among them, Fiber Bragg Gratings (FBGs) are a versatile solution, since they enable the realization of a large number of sensing points in a small space [10]. Previous studies have shown that FBG sensors can successfully operate in a very large range of temperatures, from cryogenic conditions [11] to very high temperatures [12]. Temperature measurement based on FBGs has been used in many applications, taking advantage of their very fast response time compared to more widely used electric sensors, such as thermocouples or resistance thermometers [13], as well as their high multiplexing potential [14], allowing one to have access to an increased number of measuring points with a more compact system. The fast response time also reduces the overall acquisition time, hence allowing higher acquisition rates. For in-bulk measurements in the presence of flows, optical fibers’ small size (starting from around 100 μm in diameter) is particularly attractive. In our projected application of the thermal mapping of a fluid layer, the high multiplexing potential (up to 50 sensing points in one fiber) is also an advantage, allowing one to significantly reduce the logistical requirements of the experiment.

In previous studies, the temperature was mostly measured at only a few selected positions inside the fluid or by means of ring-shaped arrangements of up to 16 sensors on the inner side wall of the convection cell (see, for example, [15,16,17]). In this way, estimations of temperature gradients or the rough reconstruction of large-scale flow structures were possible, but the resolution and dynamics of two-dimensional thermal structures could not be reconstructed from these data. The use of the aforementioned advantages of FBGs for mapping the temperature in liquid metal convection experiments in a laboratory setup and the availability of highly resolved temperature distributions inside the convection cell could be a game changer in this research field. However, the presence of the temperature sensors in the fluid volume must not significantly disturb the flow to be investigated. Optical fibers are thin and therefore very attractive for this purpose, but access with FBG sensors has a decisive disadvantage. In addition to temperature changes, these sensors are very sensitive to mechanical deformations due to tension and shape. The local flow on the fiber can also generate mechanical deformation and thus a significant signal component, which is very difficult to distinguish from the contribution of the temperature. Specific measures must therefore be taken to ensure an unimpeded determination of the temperature.

A common measure is to stiffen the fibers by coating or guiding them in a rigid housing like cladding tubes. For temperature measurements, various type of coated fibers have been developed, allowing one to increase the sensitivity and measuring range [18,19,20,21,22]. Such coatings slightly increase the fiber’s diameter, typically from 50–100 μm to a few hundred micrometers. Furthermore, they are usually encased in a rigid housing (e.g., stainless steel tubing, typically >1 mm in diameter) to filter out strain effects from the flow [23,24,25]. Since this is inevitably associated with a noticeable increase in the fiber diameter, we first looked for a possible solution using coated fibers without housing. We wished to pursue the approach of placing the fibers under a pre-tension that made them immobile so that a flow no longer caused any additional significant deformation. This paper aimed to investigate whether reliable and reproducible temperature measurements in the bulk of the fluid could be achieved by fixing the fibers under pre-tension. This procedure of the direct mounting of the optical fibers between the solid side walls for in-bulk measurements has, to our knowledge, not yet been investigated (see Yi et al. [26] for measurements with a fiber mounted at the surface of the solid wall. In this paper, we present a systematic study of temperature measurements with FBGs in a dedicated experimental setup with quiescent fluids. Each FBG sensor was paired with a standard thermocouple whose measured values were used for validation. On the basis of the results obtained, we discuss to what extent the proposed procedure is suitable for practical applications with regard to temperature measurements in liquid flows.

## 2. Description of the Problem

Fiber Bragg Grating (FBG)-based sensors use the photo-sensitivity of the core of an optical fiber to indirectly infer information about physical quantities in the environment at each sensing point (see reviews in, e.g., Othonos [27], Sahota et al. [10]). By their very nature, FBG measurements are based on the deformation of the fiber, which is caused by either temperature changes or force effects. Therefore, it is crucial to exclude the measurements from effects that are not of interest. For temperature measurements, external encapsulation is usually used to protect the fiber from strain-driven deformation effects.

The present study was motivated by our investigations into thermal liquid metal convection (e.g., [4,5]). The direct measurement of the temperature field across a designated plane within the fluid volume would be an important step towards a better understanding of the heat transport through the large-scale flow and would provide important experimental data for the validation of numerical simulations. This goal could be achieved by spanning a network of optical fibers with a large number of FBG measurement points in the fluid. These fibers should be very thin so that the flow to be measured remains unaffected by the sensors. The sheathing or coating of the fibers would increase both their diameter and the flow resistance of the sensor network. Thus, there would be a risk that the sensors could distort the flow. Conversely, the thermally driven flow, whose velocities in turbulent liquid metal convection are typically a few centimeters per second [4], would also affect the measured values by adding strain on the sensors. For this reason, the mounting of the fiber is of central importance in order to avoid distortions of the temperature measurements by the additional deformation, displacement, and vibration of the fiber caused by the fluid flow [28]. We decided on an approach in which the fiber is subjected to tensile force of a defined intensity and is firmly glued in place. This kind of fiber mounting should ensure that the measuring positions are fixed and that the temperature measurements are not affected by the flow-induced strain.

To specify the strength of the tensile force to which the fiber is exposed, we present some theoretical considerations in order to estimate to what extent the fiber diameter influences disturbances of the temperature measurements in a flow. Optical fibers are cylindrical obstacles to the flow, and the corresponding effect can be quantified by means of the Reynolds number:(1)$$Re=\frac{UD}{\nu },$$
where *U* is the flow velocity, *D* is the fiber diameter, and ν is the kinematic viscosity of the fluid.

For Re≲40, the wake created by the fiber is stable, local, and short-lived. For Re≳40, the wake generated is unstable, long-lived, and can propagate in the fluid [29]. Table 1 lists Re estimates for selected fluids of interest for laboratory experiments: air, water, and the liquid metal eutectic alloy GaInSn [30,31]. Re is calculated at the highest expected flow velocity of 10 cm/s and two different diameters, corresponding to a coated non-encapsulated fiber (D=0.2 mm) and the smallest feasible encapsulation (D=1 mm). For an encased fiber, Re is above the stability threshold for both water and GaInSn, whereas the flow around a non-encapsulated fiber remains in a region in which only small disturbances occur for water and slightly larger ones for GaInSn. This confirms that noticeably fewer flow disturbances can be expected for both fluids considered here when coated, non-encapsulated fibers are used.

The Reynolds number can be used to compute the drag force $F_{d}$ on the fiber through the drag coefficient $c_{d}$:(2)$$F_{d}=\frac{1}{2}\rho U^{2}c_{d}D,$$
where ρ is the density of the fluid.

Empirical data [32] (p. 17) show that $c_{d}$ varies steeply for our expected range of parameters, $Re≲10^{2}$. The corresponding values of $c_{d}$ for encased and coated fibers are given in Table 1. By implementing a force balance in Equation (Equation 2), the drag force can be used to estimate the tension that needs to be applied to the fiber during assembly to compensate for the flow effect. This can then be converted to a minimum equivalent mass $m_{eq}$:(3)$$m_{eq}=F_{d}/g=\frac{1}{2g}\rho U^{2}c_{d}D,$$
with *g*, the value of the gravitational acceleration, taken as 9.81 m/s^2^.

Estimations of the equivalent mass needed to keep the fiber immobile in the presence of a flow of 3 cm/s are given in Table 1, providing a guide for the experimental assembly procedure.

## 3. Experimental Setup

To determine whether accurate temperature measurements are feasible with a non-encapsulated FBG fiber mounted under tension, a benchmark comparison was required. For this purpose, we built a dedicated table-top experiment where a standard thermocouple was installed at each position of an FBG sensor. The temperature readings from the FBG were compared point by point with the readings from the thermocouples.

The experimental setup consisted of a 24 cm high column with a square cross-section of 6 × 6 cm^2^. The column was made of 5 mm thick polymethyl methacrylate (PMMA) on all sides, fixed with screws to a 1 cm thick copper plate at the bottom. The top cover made of PMMA was glued to the side walls and was used to hold the optical fiber. The top of the column also had a 1 cm diameter hole, used to insert a heating element, allowing for local heating and response time testing. For the heating, we used a 8 W electrical heater (Weller ref. WP-80, Weller, Berlin, Germany).

The top and bottom of the column were drilled with 0.5 mm diameter holes for the fiber. The fiber was coated with Ormocer^®^ and contained 24 FBGs over a length of 23 cm, with a distance of 1 cm between the individual grids, spanning from the first measuring point at z=0 to z=−22 cm. The fiber was mounted in the middle of the column with rigid epoxy glue (Weicon ref. Easy-Mix0S50, Weicon, Muenster, Germany). To ensure the required tension, one end of the fiber was first glued to the top of the column and left to cure. When the glue had fully solidified, a weight of 50 grams was attached at the bottom end of the fiber with a piece of tape, and the fiber was glued to the bottom plate. The weight was removed once the glue was fully solid. The attached weight was well above the minimum equivalent mass $m_{eq}$ needed to balance the expected drag force, as estimated in Equation (Equation 3). During gluing, special attention was paid to ensure correct alignment between the FBGs and the thermocouple holes on the side wall. The setup, including all components, is shown in Figure 1.

The thermocouples were inserted from the side wall and positioned so that the tip of each sensor was as close to the fiber as possible; see insert in Figure 1. Thermocouples of type K were glued in place with flexible silicon (Weicon ref. Flex 310M). In total, there were 23 thermocouples in the side wall, each corresponding to an individual FBG, the last FBG being in the bottom plate.

The fiber was connected to a broadband spectrometer (FiSens ref. FiSpec FBGX152, FiSens, Braunschweig, Germany) with an emitting spectrum ranging from 795 nm to 885 nm. The FBGs were each assigned a reference wavelength by the manufacturer, from 822.5 nm for the first one at z=0 to 880 nm, and spaced every 2.5 nm, ensuring that the spectrometer spectrum encompassed the FBGs’ reflections. The optical fiber’s temperature variation coefficient was $8.65\times 10^{-6}$ K^−1^, according to the manufacturer. An example of the reflected spectrum is shown in Figure 2. The wavelength data were read and converted into temperature with the proprietary software of Fisens (FiSens ref. BraggSens v. 1.81). The reflected wavelength for each FBG was extracted from the reflected light signal using Gaussian peak detection [33]. We assigned fitting windows of 2.5 nm centered on the reference wavelength of the FBG and obtained the reflected wavelength from the fit results. The difference between the reference and measured wavelength allowed us to deduct the temperature difference thanks to the temperature coefficient; in our case, the displacement was typically below 0.1 nm.

To ensure the high precision of the thermocouples, they were connected to an external temperature reference (Klasmeier ref. Isotech TRU 937/50, Klasmeier, Fulda, Germany), which, along with custom calibration, ensured a precision of 0.03 K. Thermocouple measurements were read by a digital multimeter (Keithley ref. DAQ6510, Keithley, Cleveland, OH, USA) through multiplexer cards (Keithley ref. 7710) with an acquisition time of 1 s. Both types of temperature data, from the FBGs and thermocouples, were post-processed and compared using a Python script. An overview of the data acquisition process is shown in the flowchart in Figure 3.

## 4. Results

### 4.1. Method

As shown in the previous sections, the fiber was mounted under tension and was subject to deformation, which arose from both the strain and temperature. In order to acquire accurate temperature measurements, the contribution from strain had to be taken into account and compensated for. When FBG sensors are used in flows, the deformation of the fiber depends on the local velocity and temperature variations of the fluid, which makes the prediction of the strain effect and thus correction very difficult. Therefore, we monitored the tension by a reference sensor and used this reference value for ad hoc corrections to remove tension effects at each time step and extract an accurate temperature from the FBG. For this purpose we used one FBG sensor and assumed that the measured strain was representative for the entire fiber. This difference correction method can be implemented with the lowest error when the reference strain FBG is exposed to minor temperature variations or at best kept at a constant temperature. In practice, one can customize the sensing fiber to position the reference FBGs far from the region of interest, or outside of the frame (e.g., in the wall of the enclosing structure).

The sensor signal *A* at a given position *i*, denoted $A_{i}$, is composed of a temperature component $T_{i}$ and a strain component $S_{i}$: $A_{i}=T_{i}+S_{i}$. We assume that the strain component is constant over the entire fiber $S_{i}=S_{ref}$ and can therefore be measured at a reference point $S_{ref}$ and subtracted in a simple way. To determine the absolute temperature, it is necessary to measure the temperature at the reference measuring point ($T_{ref}$) using a thermocouple. This gives the required temperature value at the measuring point *i*: $T_{i}=T_{ref}+\left(A_{i}-A_{ref}\right)$, with $A_{ref}$ being the sensor signal at the reference position. This method relies on the difference from a reference; therefore, it only allows for relative temperature measurements:(4)$$T_{diff\;i}^{\mathrm{FBG}}=T_{i}^{\mathrm{FBG}}-T_{ref}^{\mathrm{FBG}}$$
with $T_{ref}^{\mathrm{FBG}}$ being the apparent temperature from the *FBG* chosen as the strain reference, $T_{i}^{\mathrm{FBG}}$ the temperature from the *FBG* at the measuring position *i*, and $T_{diff\;i}^{\mathrm{FBG}}$ the deducted relative temperature at the position *i*.

A solution to obtaining absolute temperature measurements is to add one thermocouple (or any stable calibrated temperature sensor) to the setup, close to the chosen strain reference FBG. In practice, it is often acceptable to add one sensor in a less disturbing location for the flow while minimizing additional costs and logistics. For thermal convection experiments, a suitable position for such a reference sensor is close to, or embedded in, one of the enclosing walls of the setup. The absolute temperature $T_{abs}^{\mathrm{FBG}}$ can then be retrieved by adding the thermocouple’s reading at the reference point $T_{ref}^{\mathrm{TC}}$ to the relative variation in temperature of the FBG obtained by the above-described difference method:(5)$$T_{abs}^{\mathrm{FBG}}=T_{diff\;i}^{\mathrm{FBG}}+T_{ref}^{\mathrm{TC}}.$$

In the testing setup presented in this study, a well-suited position would typically be close to either the top or the bottom of the fluid column. This ad hoc compensation is similar to the method presented by [34] for solid surface measurement, but we kept the measurement system more versatile by not adding differentiated encapsulation for strain and temperature FBGs.

In the next sections, we present two example cases of temperature mapping using FBGs. We first show that we were able to reconstruct a stationary temperature profile in the liquid metal GaInSn. We then show that time-dependent temperature variations could be tracked by including a local time-dependent heat source. For technical reasons, this second case was conducted in water.

### 4.2. Stable Temperature Gradient in GaInSn

The experimental column was filled with GaInSn, and the copper base of the column was connected to a cold water bath with a constant temperature of 13 °C (the ambient temperature was 21 °C). This caused a temperature gradient to build up in the fluid. As reference for the strain, we chose the FBG at the top of the column z=0, where the fluid temperature was almost identical to the ambient temperature, and we applied the correction given in Equation (Equation 4) to all the other sensor positions. Since we had access to the thermocouples across the whole length of the column, we started the data acquisition before thermal equilibrium was reached, which also gave us an impression of how the temperature gradient built up in the fluid.

Figure 4 shows the reconstructed vertical profile of the temperature for two time steps (100 s and 1000 s) after initiating the cooling process. The thermal relaxation time for this configuration was approximately 4000 s. The temperature profiles delivered by the FBGs (solid lines) were compared to the benchmark data provided by the thermocouples (dashed lines). The temperature profiles revealed a clear gradient across the vertical positions, showing that spatial temperature variations were successfully measured from the FBGs along the single fiber. For the upper half of the column, both the gradient profile and amplitude measured by the optical fiber agreed well with the results recorded by the thermocouples. To quantify this comparison, we defined the measurement error as the difference between the temperature measured by the FBG and extracted from our correction procedure and the thermocouple data. We plotted this quantity as shown in Figure 4b, and it became obvious that the error was very low in the first half of the column, confirming the accuracy of the fiber measurements. However, we could see that on the bottom of the column, starting from z=−14 cm, the fiber measurements started to significantly diverge from the benchmark values, and the error increased with an increasing distance from the reference sensor at the top of the fiber. It seems that in this case, the applied correction was not able to compensate for the influence of strain over the entire fiber, particularly at points far away from the reference. For comparison, the same correction was applied with the reference point at other positions of the fiber: at the bottom of the column, z=−22 cm, and at half-height, z=−14 cm. However, this variation did not solve the problem. The reference point initially chosen was able to accurately maintain the temperature profile in its vicinity, while significant deviations from the benchmark data occurred at greater distances from the reference point. We observed that choosing a reference point with a stable temperature (here, at the top of the fiber) was the best choice with the smallest deviations.

It should be mentioned that the overall temperature gradient applied in this example case was considerably larger than the expected thermal structures in a typical convection setup. The thermal structures in Rayleigh–Bénard convection are typically organized on smaller length scales and show a highly transient behavior. In this respect, it made sense to perform another experiment to prove the reliability of the method in the case of more similar thermal structures. The second example deals with a local heat source when the setup was filled with water.

### 4.3. Local Time-Dependent Heating in Water

We used the same experimental setup, now filled with water as the working fluid, and introduced local electrical heating from the top of the column (see Figure 1). The electrical heater was connected to a power source at a constant voltage that input a constant heating power into the system and allowed precise time synchronization. The measurement started when the setup was at thermal equilibrium at room temperature. The electrical heater was turned on at t=250 s and turned off at t=650 s. The temperature time series is shown in Figure 5.

As a reference point for the strain measurement, we chose the sensor at the bottom of the column, furthest away from the heat source, at z=−24 cm. Time series of the corrected temperature $T_{diff}$ are shown in Figure 5 for the positions z=0 to 13 cm (the positions further away from the heat source showed a constant temperature) as solid lines for the FBG measurements in comparison with the thermocouple data as dashed lines. The FBG results were in good agreement with the benchmark data, and the general evolution and amplitude were well captured at each position over time. Differences between the two datasets became obvious after changes in the heating power. The measurements diverged immediately after the heating was turned on (or off), while the response of the FBG sensors to the temperature changes occurred significantly faster than that of the thermocouple measurements, especially for the positions close to the heater (z=0 to −3 cm).

This effect was attributed to the acquisition time of the respective sensors, due to both the thermal response time inherent to the sensor and the different acquisition rates of the two measuring systems. The response time was about 200 ms for the thermocouples and 50 ms for the FBG sensors (given by the manufacturer). Moreover, the acquisition rates were very different due to the nature of the system: the thermocouples were multiplexed by analog switching (the sensors were read one after another), whereas the FBGs were read all at the same time thanks to the broadband spectrometer. The effective acquisition frequency was then 1Hz for the thermocouples and 18 Hz for the optical fiber, which, combined with the sensor’s thermal response time, explains the delay we observed in Figure 4. Thus, the FBG technique offers a significant increase in temporal resolution, which is an advantage for the detection of fast-moving thermal structures.

Figure 6 shows vertical temperature profiles at four time steps, which are indicated by the vertical lines in Figure 5: before heating (in purple), immediately after the start of heating (blue), shortly before turning off heating (green), and after some relaxation time (yellow). All profiles present clearly distinct features, proving that the transient behavior of the local thermal anomalies could be measured and tracked in time with one optical fiber in the bulk. We remark that some discrepancies existed between the FBG and thermocouple data, which could be explained by the difference in time response. The most obvious discrepancy could be seen shortly after heating was applied (in blue): the FBGs (solid line) showed an increase in temperature, indicating that the optical fiber was already sensitive to the local heating, while the thermocouple temperature was still constant (dashed line).

The absolute temperature measurement could be determined if one thermocouple was taken as reference. For this case, we chose the thermocouple corresponding to the FBG strain reference, at the bottom of the fluid column. In our example, the temperature at the reference position was almost constant at a value of 22.81 ± 0.03 °C. This allowed us to take the average value of the time series and add it as a constant offset, following Equation (Equation 5). The deducted absolute temperature values are shown in the right axis of Figure 5 and the top axis of Figure 6. It can be noticed that some discrepancies existed after ad hoc strain correction, which can be seen in Figure 5, where the temperature readings between the thermocouples and FBGs could differ by up to 0.5 K. Part of this deviation could be interpreted as an effect of the difference in response time, as mentioned above. The FBG measuring system relying on a coated optical fiber thereby repeatedly demonstrated its superiority in terms of time resolution. Therefore, such a system is recommended for applications of relative temperature measurements, where a high accuracy in absolute temperature is not the main focus.

## 5. Discussion

In this study, we considered whether temperature measurements using FBG fibers in a flowing fluid are feasible. An essential question was whether it was possible to design the measurement system in such a way that the temperature data were not distorted by local deformations of the fiber due to transient flow effects. FBG sensors are inherently sensitive to deformation, and the influence of flow-driven mechanical forces needed to be taken into account for acquiring reliable temperature data. We pursued the idea of applying a defined pre-tension to the fiber, which stiffened it and thus prevented flow-induced deformation. This approach was tested for its suitability in a dedicated experimental setup where stagnant fluids such as the ternary metal alloy GaInSn and water were exposed to a temperature gradient and local heating, respectively. The results provided by the FBG sensors were verified by the temperature data obtained using standard thermocouples, which were installed in the immediate vicinity of the FBG positions. We used an optical fiber coated with Ormocer^®^, mounted directly without encapsulation, in order to achieve the thinnest possible cross-section of the fiber (∼200 μm), thus limiting the disturbances to the flow due to the presence of the sensor to a minimum. The assumption that omitting a sheath from the fiber would noticeably reduce the transient character of the wake and thus the changes in the flow to be measured was supported by theoretical predictions. Estimations of the drag force expected on the fiber allowed us to define a protocol for mounting the fiber so that it stayed immobile in the presence of an acting flow. In practice, the fiber was glued under a pre-tension that was significantly larger than the drag force exerted by the expected fluid flows.

We suggested a post-processing method based on ad hoc strain correction by taking one FBG as the strain reference and using its readings to correct the measurements from the other FBGs located within the same fiber. This method is considered to be straight-forward and has low effort and logistics requirements for implementation, as it employs one FBG sensor for correction that cannot be used for temperature measurements without adding specialized sensors. This approach was tested for reliability using two example cases, a stable thermal gradient in liquid metal (GaInSn) and a local time-dependent heat source in water. We demonstrated that in both cases, the optical fiber sensors were able to measure and follow temperature variations in both time and space, in good agreement with the benchmark provided by thermocouples. The measurement of absolute temperature values could be achieved by adding a stable reliable temperature sensor (e.g., thermocouple) near the FBG strain reference position. Due to the small size of the optical fiber sensors, their response time was faster than that of the benchmark sensors.

However, the agreement between the temperature values of the FBG sensors and the measurements of the thermocouples was only satisfactory over approximately half the length of the examined fiber, starting from the reference point for the tensile stress measurement. This discrepancy apparently increased with an increasing distance from the reference point. This is a clear indication that the proposed method cannot be used for arbitrarily long fibers and only for a finite number of neighboring FBGs.

We assume that this problem can be explained as follows. If the fiber is firmly glued between the side walls of the fluid container under pre-tension, not only is tension imposed, but the length of the fiber is also fixed. If this fiber is now placed in a temperature gradient, different thermal expansions occur along the fiber. If the fluid container is cooled at the bottom as in the first experiment, the FBGs measure not only the local thermal expansion, but also an additional tensile stress that results from the entire fiber contracting due to cooling. In our case, however, only the tensile stress determined by the upper reference sensor was compensated for, which was located at the warmest location. This effect was not sufficiently taken into account in the design of the measurement system. Remedial action can be taken through the following measures: (i) The fiber is placed under pre-tension, but only fixed to one wall. In this way, one would retain a pre-tension that avoids the falsification of the measured value by the flow and allow an additional change in the length of the fiber due to thermal expansion. However, this variant would be difficult to implement in practice, as the side on which the fiber is free to move would be difficult to seal against fluid leakage. (ii) Another possibility is to use more FBGs as reference sensors for a tension measurement. However, this would mean losing temperature measurement points. It may be possible to compensate for this by laying two fibers directly next to each other and measuring temperature and tensile stress alternately. (iii) A slight influence on the flow is accepted by encasing the fiber in a sheath. This approach should work best for small Rayleigh numbers and low flow velocities.

In summary, for the characterization and tracking of thermal structures where relative variations are paramount, a coated, non-encapsulated optical fiber mounted under tension with multiple FBG positions is a promising solution as it offers a fast response time and high sensitivity while providing a sensor with a very small diameter and high multiplexing capabilities for minimal flow disturbances. However, it must be taken into account that the effective compensation of the influence of the tensile stress is necessary to obtain accurate temperature data.

For future applications, careful considerations need to be made for each specific experimental system and problem of interest, as a compromise between the acceptable disturbance to the flow and the precision of temperature data has to be reached. Typically, for experiments where temperature measurement with high accuracy is targeted, an encased fiber is preferable. This still retains many of the advantages of optical fiber technology, such as multiplexing, passiveness, and a large temperature range. For turbulent thermal convection experiments, large-scale mapping in the bulk may be more advantageous with an encased fiber, provided that the housing is kept at a minimal size, since the absolute temperature can provide insightful information about heat transport. For local in-bulk measurements, typically focusing on structures like plumes and boundary layers, where the integrity of the structures is crucial, a coated fiber without encapsulation, as presented in this study, offers an interesting alternative.

## Figures and Tables

**Figure 1 sensors-23-08539-f001:**
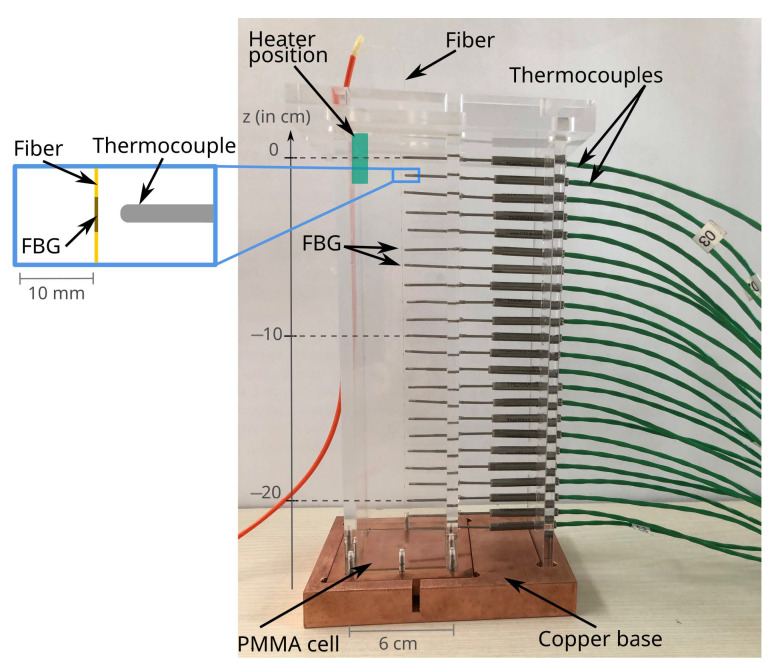
Side-view photo of the experimental setup with its instrumentation: optical fiber in the center of the column with 24 FBGs spaced every 1 cm, 23 thermocouples with sensing points corresponding to the FBG positions. The fiber was encased outside of the measurement area in an orange cable connected to the spectrometer, and the measuring area of the coated fiber was positioned vertically in the middle of the setup, barely visible in the picture (see left insert for detailed schematic). The heater was later inserted from the top, schematized in green.

**Figure 2 sensors-23-08539-f002:**
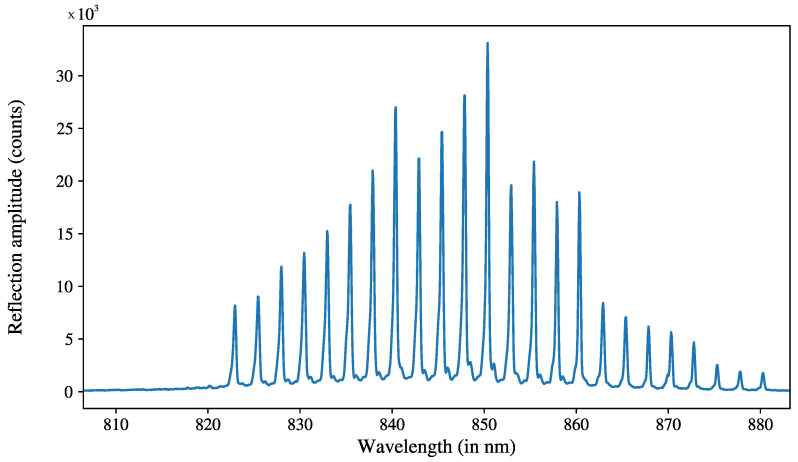
Example of reflected spectrum from the spectrometer. Each FBG reflected a slightly different wavelength, enabling the easy multiplexing of the sensors. The FBG at position z=0 showed a reference reflective wavelength of 822.5 nm. The temperature differences to be measured here caused shifts in the wavelength within 0.1 nm.

**Figure 3 sensors-23-08539-f003:**
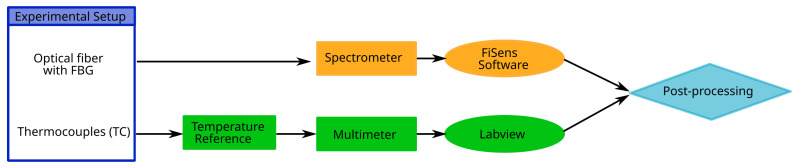
Flow chart of data acquisition procedure.

**Figure 4 sensors-23-08539-f004:**
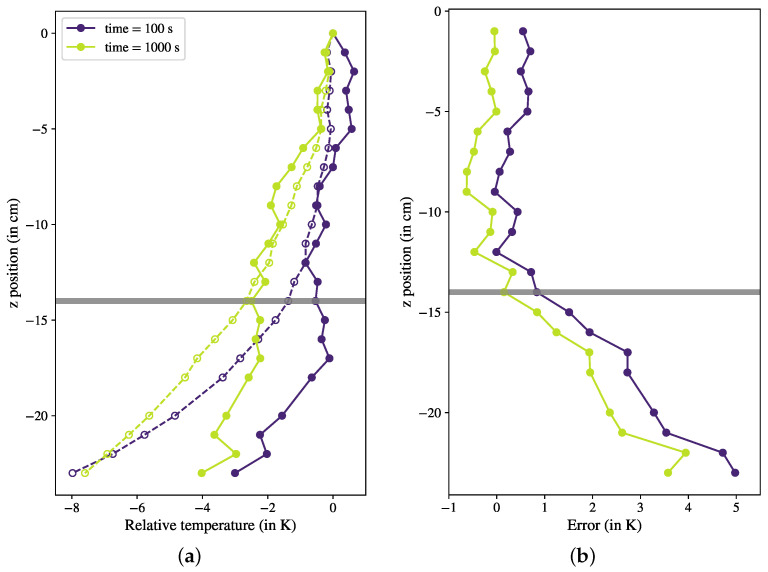
(**a**) Reconstructed temperature profile for stable gradient in GaInSn. Profiles are shown for two snapshots of FGB sensors (solid lines, closed circles) and compared to thermocouple measurements (dashed lines, open circles). (**b**) Measurement error, defined as the difference between FBG and thermocouple measurements, taken as benchmark.

**Figure 5 sensors-23-08539-f005:**
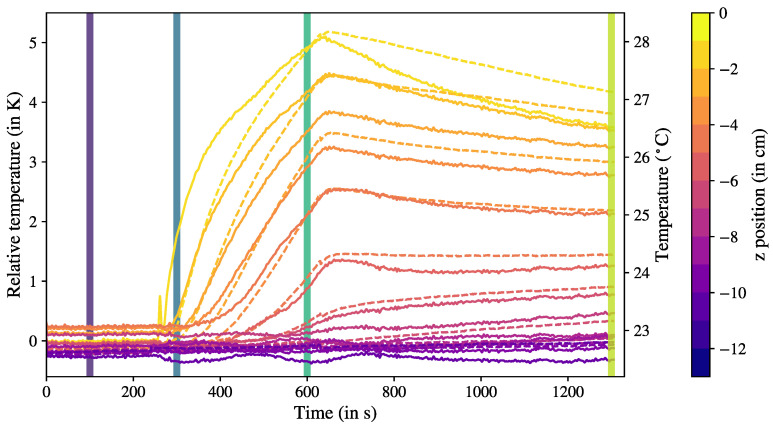
Temperature time evolution with local heating of thermocouples (dashed lines) compared with FBG readings (solid lines) corrected with respect to reference sensor. Left axis gives the relative temperature, right axis gives the absolute scale by taking the reference temperature as the time-average of the thermocouple data at the reference sensor, at z=−24 cm. Vertical solid lines corresponds to the times at which temperature profiles were computed in Figure 6, from purple to green at 100 s, 300 s, 600 s, and 1300 s.

**Figure 6 sensors-23-08539-f006:**
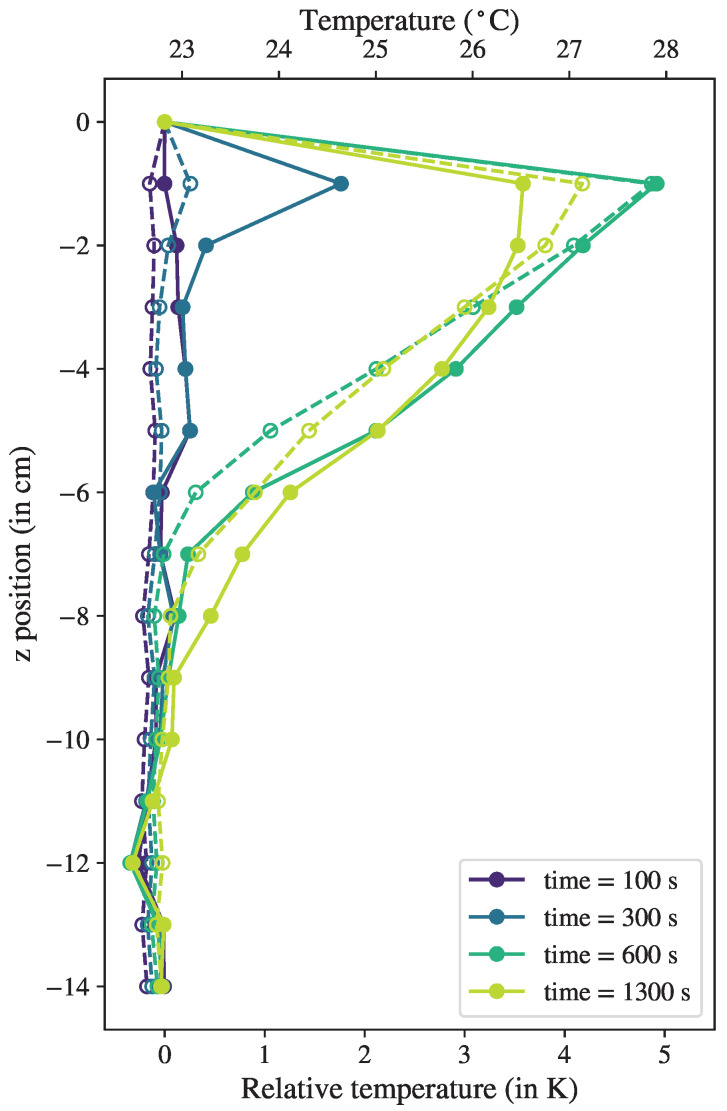
Reconstructed temperature profiles at four time steps: t=100 s, 300 s, 600 s, and 1300 s. Profiles from optical fiber (solid lines) are compared with thermocouple benchmark (dashed lines). The colors of the profiles correspond to the vertical lines in matching colors from Figure 5.

**Table 1 sensors-23-08539-t001:** Estimations of Reynolds number, drag coefficient $c_{d}$, and equivalent needed mass $m_{eq}$ for air, water, and GaInSn considering fibers of two diameters: D=1 mm (encased fiber) and D=0.2 mm (coated fiber). Quantities were computed for a flow velocity of U=10 cm/s.

		D=1 mm	D=0.2 mm
	Re	7	1
Air	$c_{d}$	3	10
	$m_{eq}$ (g)	$2\times 10^{-3}$	12
	Re	100	20
Water	$c_{d}$	2	2
	$m_{eq}$ (g)	1	0.2
	Re	300	60
GaInSn	$c_{d}$	1	2
	$m_{eq}$ (g)	3.2	1.3

## Data Availability

Data supporting this study are available at https://doi.org/10.14278/rodare.2463 (accessed on 20 August 2023).

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
