# Peer review of "In-Bulk Temperature Profile Mapping Using Fiber Bragg Grating in Fluids"

_sensors, 2023, doi:10.3390/s23208539_

Round 1

Reviewer 1 Report

This manuscript presents a compelling study examining the performance of Fiber Bragg Grating (FBG) sensors in mapping temperature profiles within bulk materials. The authors successfully achieved accurate temperature measurements by implementing a strain reference correction in both case studies. This contribution significantly enriches the existing literature on the subject and merits publication. In the following sections, I provide some corrections and comments aimed at improving the clarity of the text.

(1)   Page 7, line 219. ‘The applied correction is not able to compensate the influence of strain over the entire fibre, in particular far away from the reference’. It raises questions regarding the uniformity of deformation/tension across all FBG sensors. It might be worthwhile to consider the thermal performance of the entire experimental setup, including thermal boundary conditions at both the bottom and top of the system. Additionally, providing information on the 'ground truth' of the temperature profiles would enhance the context.

(2)   Fig. 4. The colour bar on the right-hand side may be misleading. Since the Z position at 0 cm represents the top of the system and the temperature is rising due to heating from the top, the colour bar should reflect this with warmer colours at the top and cooler colours at the bottom.

(3)   Line 244. ‘Both these effects are directly linked to the response time of the sensors, which can be easily understood by comparing both sensors’ size: optical fibre 200 μm in diameter whereas the thermocouples are 2mm, causing the response time to be faster in the optical sensors ’. It is conceivable that the optical fibre sensors, with a diameter of 200 μm, may exhibit a faster response time compared to the thermocouples, which have a larger diameter of 2 mm. However, it would be beneficial to provide empirical evidence supporting this claim, particularly with regard to any observed time differences.  

(4)   Line 262. ‘following 5.’ It's not entirely clear what "following 5" refers to without further context. Please consider providing additional information or rephrasing for clarity.

This manuscript presents a compelling study examining the performance of Fiber Bragg Grating (FBG) sensors in mapping temperature profiles within bulk materials. The authors successfully achieved accurate temperature measurements by implementing a strain reference correction in both case studies. This contribution significantly enriches the existing literature on the subject and merits publication. In the following sections, I provide some corrections and comments aimed at improving the clarity of the text.

(1)   Page 7, line 219. ‘The applied correction is not able to compensate the influence of strain over the entire fibre, in particular far away from the reference’. It raises questions regarding the uniformity of deformation/tension across all FBG sensors. It might be worthwhile to consider the thermal performance of the entire experimental setup, including thermal boundary conditions at both the bottom and top of the system. Additionally, providing information on the 'ground truth' of the temperature profiles would enhance the context.

(2)   Fig. 4. The colour bar on the right-hand side may be misleading. Since the Z position at 0 cm represents the top of the system and the temperature is rising due to heating from the top, the colour bar should reflect this with warmer colours at the top and cooler colours at the bottom.

(3)   Line 244. ‘Both these effects are directly linked to the response time of the sensors, which can be easily understood by comparing both sensors’ size: optical fibre 200 μm in diameter whereas the thermocouples are 2mm, causing the response time to be faster in the optical sensors ’. It is conceivable that the optical fibre sensors, with a diameter of 200 μm, may exhibit a faster response time compared to the thermocouples, which have a larger diameter of 2 mm. However, it would be beneficial to provide empirical evidence supporting this claim, particularly with regard to any observed time differences.  

(4)   Line 262. ‘following 5.’ It's not entirely clear what "following 5" refers to without further context. Please consider providing additional information or rephrasing for clarity.

Author Response

We thank the Referee for the careful reading and reviewing of our manuscript and for helpful remarks. We have tried to follow all suggestions and made corresponding corrections in the text. The changes that we have made in the revised version of the manuscript are marked in red. We respond to the comments of the Referee one by one in the attached pdf.

Reviewer 2 Report

The authors claimed that they have proposed In-bulk temperature profile mapping using Fiber Bragg Grating in fluids. This is a significant work, but before publication authors need to clarify some points given below:

1. First of all, the authors are suggested to improve the descriptions of the abstract and highlight the impact of this work.

2. Please explain the meaning of Re in line 93 and Table 1.

3. The explanation in Figure 1 is not clear. Please provide a detailed explanation of the relationship between the various parts in Figure 1, such as whether each FBG is equipped with one thermocouple; What is the relationship between the three parts from bottom to top? FBG serial labeling is not obvious, etc.

4. Please explain the meaning of each variable in eq. 4 and 5.

5. Figure 4 shows an unclear representation, especially the relationship between the curve and the two vertical coordinates is even more unclear.

6. The paper used Sheathing or coating FBG for temperature measurement, but the research results lacked experiments and analysis on the impact of fluid disturbance on the measurement results, and there was no proposed method to remove the influence of disturbance.

7. Please add a comparison of the research results with other studies to illustrate the advantages of this study.

Author Response

(The authors gave the same response as above.)

Reviewer 3 Report

The work was done at a high scientific level: the theory is rather succinctly presented, but understandable. The experimental data are well presented, I would like to note separately that even the probable errors in measurements are presented. This work deserves publication in the journal "Sensors" after minor additions:

1. Could you briefly characterize the FBG: describe in more detail the physical parameters, the manufacturing method, give the transmission or reflection spectra

2. Describe in  more detail the method of measuring temperature using FBG - give the displacement spectrum for at least one point - as an example.

3. Can fluid flows somehow affect fiber tension?

Author Response

(The authors gave the same response as above.)

Round 2

Reviewer 2 Report

After the revision, this paper can be accepted, in my opinion.

Author Response

We thank the referee for the kind feedback.